# Knowledge, attitude, and practice on blood donation among undergraduate first-year engineering students in Nepal

Bhola Teli[1,2*], Durga Rijal[1,2], Sulochan GC[3], Ashok Khanal[3], Anil Kumar Singh[2]

1 B.P. Koirala Institute of Health Sciences (BPKIHS), Dharan, Nepal, 2 Central Department of Public Health (CDPH), Institute of Medicine (IOM), Tribhuvan University (TU), Kathmandu, Nepal, 3 University of South Carolina Arnold School of Public Health, Columbia, South Carolina, United States of America

* bholateli23@gmail.com

## Abstract

### Background

Blood donation is essential for healthcare, yet maintaining a safe supply remains a challenge in low- and middle-income countries like Nepal. While university students are a key donor demographic, data on engineering students—a group with distinct academic profiles—is limited. This study assessed the knowledge, attitudes, and practices (KAP) regarding blood donation and identified the associated factors of these domains among first-year engineering students in Nepal.

### Methods

A cross-sectional study was conducted between April and June 2022 among 191 students at the Pulchowk Engineering Campus, Tribhuvan University. Data were collected via a structured online questionnaire. Associated factors of knowledge and attitude (continuous scores) were identified using multivariable linear regression, while associated factors of donation practice were analyzed using multivariable logistic regression. Internal consistency was assessed using Cronbach's alpha.

### Results

The mean knowledge score was 13.91 (±3.81) out of 23, and the median attitude score was 7.00 (IQR: 1.00). Although attitudes were highly favorable, actual donation practice was low (16.8%). Multivariable linear regression showed that male students had significantly lower knowledge scores than females (B = −1.26, p = 0.048), and knowledge was the sole significant associated factor with positive attitudes (B = 0.04, p = 0.025). Logistic regression revealed that age ≥ 20 years (OR = 2.83, p = 0.030) and being male (OR = 4.20, p = 0.029) were strongly associated factors of donation practice, while civil engineering students were less likely to donate than those in other

**Data availability statement:** All relevant data are within the paper and its Supporting information files.

**Funding:** The author(s) received no specific funding for this work.

**Competing interests:** The authors have declared that no competing interests exist.

programs (OR = 0.38, p = 0.024). The primary barriers reported were "no specific reason" and "lack of opportunity."

## Conclusion

A descriptive "knowledge-practice gap" exists among engineering students. While females are more knowledgeable, males are more likely to donate. Interventions should move beyond general awareness toward structural mobilization, such as regular on-campus donation drives, to convert high ideological support into active practice.

## Introduction

Blood is an essential body fluid that transports nutrients, hormones, immune factors, and waste products throughout the body [1,2]. Blood products, including whole blood, blood components, and plasma-derived medicinal products, are critical for the treatment of patients with medical conditions, injuries, bleeding disorders, and surgical needs [3]. Blood donation, the voluntary act of giving blood for medical use, is therefore a vital component of healthcare systems worldwide [4,5]. It helps reduce maternal and child mortality and supports the treatment of conditions such as haemophilia, thalassaemia, immunodeficiencies, cancer, trauma, and complications requiring major surgery or organ transplantation [3].

Blood donation may involve whole blood, red blood cells, platelets, or plasma, and may be classified as allogeneic or autologous. Based on donor type, donations are categorized as voluntary non-remunerated, family replacement, or paid donation [6,7]. Among these, voluntary non-remunerated blood donation is considered the safest and most sustainable source because it has the lowest risk of transfusion-transmissible infections [6].

Despite its importance, blood donation remains insufficient in many low- and middle-income countries. Globally, 118.5 million blood donations were collected in 2018, but distribution was uneven across regions and income groups [8]. High-income countries contributed a disproportionate share of donations compared to their population size, while lower-middle-income countries contributed less despite having a much larger share of the global population [8]. In Nepal, 242,048 blood donations were recorded, which is below the World Health Organization's recommended minimum coverage of 1% of the population [9–11]. At the same time, the supply of blood to hospitals exceeded collection, indicating a significant gap between demand and availability [10]. This shortage has contributed to delays in treatment, particularly in emergency care and surgery, and in some cases has led to preventable deaths [12–15].

Young people, particularly college students, represent an important target group for improving blood donation rates in Nepal. They constitute a substantial proportion of the population, are generally healthy and socially active, and are well-positioned to influence their peers through community engagement. The World Health Organization also recognizes youth as a key group for promoting voluntary blood donation through awareness and education [9]. In Nepal, the Nepal Red Cross Society

has been entrusted with a central role in managing blood donation programs and mobilizing young people to help meet national blood requirements. However, evidence regarding blood donation-related knowledge, attitudes, and practices among engineering students in Nepal remains limited.

Previous studies conducted in Nepal and other countries have shown that, although students often demonstrate positive attitudes toward blood donation, their actual donation practice remains low [16–21]. Medical students generally exhibit better knowledge and higher rates of blood donation than non-medical students, including engineering students [16–19]. Common barriers to donation include fear, medical ineligibility, lack of awareness, and not being approached to donate [20,21]. These findings suggest that students could serve as a valuable pool of voluntary blood donors if they are provided with adequate knowledge, motivation, and opportunities to donate.

Despite the increasing demand for safe blood, blood donation remains inadequate in Nepal [10,13]. Young people, particularly university students, represent a potential source of voluntary blood donors. Although several studies have examined blood donation among medical and health science students [16,17], evidence among engineering students in Nepal is scarce. Since knowledge and attitude are important determinants of blood donation practice, understanding these factors among engineering students is essential. Therefore, this study was conducted to assess the knowledge, attitude, and practice regarding blood donation among first-year undergraduate engineering students at Pulchowk Engineering Campus and to generate evidence for interventions aimed at promoting voluntary blood donation among young adults.

## Methods

### Setting

This study was conducted at the Pulchowk Engineering Campus, Institute of Engineering (IOE), Tribhuvan University, Nepal. The campus is the central institution overseeing engineering education under Tribhuvan University and is located in the capital city, Kathmandu. Established in 1972, the campus offers bachelor's and postgraduate programs in various engineering disciplines. Each year, it enrolls approximately 576 undergraduate students across multiple programs, including Architecture, Civil, Computer, Mechanical, Electrical, Aerospace, Chemical, and Electronics, Communication, and Information Engineering.

A descriptive cross-sectional study design was adopted and conducted between April and June 2022. The study was reported in accordance with the Strengthening the Reporting of Observational Studies in Epidemiology (STROBE) guidelines (S1 File).

### Sample size determination

The sample size (n) was calculated using a single population proportion formula. The margin of error (d) was set at 5% (d = 0.05), and the prevalence of blood donation practice was taken as 20.8% based on a previous study among a similar population [22]. A 95% confidence interval ($Z\alpha/_2$ = 1.96) was applied to estimate the minimum sample size.

$$n_0 = \frac{z_{\frac{a}{2}} * p * (1 - P)}{d^2} = \frac{1.96 * 1.96 * 0.208 * 0.792}{0.05 * 0.05} = 253$$

After adjusting for the finite population:

$$n = \frac{n_0}{1 + \frac{n-1}{N}} = \frac{253}{1 + \frac{253-1}{576}} = 176$$

Considering the maximum non-response rate of 61% commonly observed in web-based studies [23], the adjusted sample size was:

$$n_{-final} = \frac{176}{0.39} = 451$$

However, to enhance the study's robustness, the entire population of 576 students was invited to participate. Achieved sample size (n = 191) exceeds unadjusted target (n = 176).

## Data collection

Data were collected using a self-administered structured questionnaire prepared in English. The questionnaire comprised four sections: Socio-demographic characteristics (age, sex, religion, ethnicity, living status, type of engineering, father's and mother's education, and residential location); Knowledge (19 items); Attitude (7 items); Practice (7 items). The questionnaire used in this study was developed specifically for this research through an extensive literature review, peer, and expert consultation [21,22,24,25]. The correct responses for the knowledge, attitudes, and practices (KAP) items in the questionnaire were derived from a review of existing literature and established national guidelines [21,22,24–26]. The questionnaire was pretested among first-year undergraduate students of the Central Department of Public Health (CDPH), Institute of Medicine, Tribhuvan University, to ensure clarity, completeness, and comprehensibility. Revisions were made based on feedback before final administration. The English version of the questionnaire is provided as a supplementary file (S2 File).

The final version was converted into a Google Form and distributed via email to all first-year undergraduate engineering students at Pulchowk Campus. Participation was voluntary. Students who did not provide consent, declined to participate, or lacked internet access during the data collection period were excluded. Data collection (participant recruitment) took place from April 28, 2022, to May 20, 2022. Confidentiality was strictly maintained, and voluntary participation was emphasized to minimize social desirability and response bias.

## Operational definitions

The variables in this study were operationally defined as follows to facilitate statistical analysis:

Knowledge Score: Treated as a continuous variable (maximum score: 23) based on 19 items. For sensitivity analysis, a score at or above the 50th percentile (median) was categorized as "Adequate Knowledge," while scores below were categorized as "Inadequate Knowledge".

Attitude Score: Treated as a continuous variable (maximum score: 7). For sensitivity analysis, scores at or above the 50th percentile (median) were categorized as a "Positive Attitude," while lower scores were considered a "Negative Attitude".

Practice: This was treated as a binary outcome based on the item "Have you donated blood before?" Responses were categorized as "Yes" (at least once) or "No" (never donated).

Age: This was treated as a categorical variable, dichotomized into "Less than 20 years" and "20 years or more" based on the sample distribution, previous study, and avoiding the linearity assumption unsupported by behavioral data [27,28].

Sex: This was categorized based on self-identification as "Male" or "Female".

Religion: Participants were categorized into "Hindu" and "Non-Hindu" (comprising Buddhist, Christian, Kirat, and Muslim participants).

Ethnicity: Participants were grouped into "Brahmin/Chhetri" and "Non-Brahmin/Chhetri" (including Janjati, Madhesi, Dalit, and other minority groups).

Living Arrangement: This was dichotomized into "Both parents" (referring to a nuclear family setting) and "Non-nuclear arrangement" (including those living alone, with a single parent, or with relatives).

Academic Programme: Participants were categorized as being in the "Civil Engineering" programme or "Other engineering streams" (includes mechanical engineering, Electronics, Communication, & Information Engineering, Computer Engineering, Electrical Engineering, Aerospace Engineering, Chemical Engineering, and Bachelor of Architecture).

Parental Education: Both paternal and maternal education levels were categorized into "Basic or below" (no formal schooling through grade 8) and "Secondary or above" (grade 9 through university level).

Family Residency: Based on administrative divisions, residency was categorized as "Municipality" or "Metropolitan City".

## Statistical analysis

The survey utilized mandatory response fields to prevent missing data; consequently, only complete entries from the 191 respondents (33.2% response rate) were included in the final analysis without the need for sampling weights. Completed responses were retrieved and verified for completeness and accuracy. Statistical procedures were performed using IBM SPSS Statistics version 27.0. Descriptive statistics, including frequencies, percentages, mean, standard deviation, medians, and interquartile ranges (IQR), were calculated to summarize socio-demographic characteristics and the levels of knowledge, attitude, and practice among the study population.

The primary analysis treated knowledge and attitude scores as continuous variables. Data normality was assessed using the Shapiro-Wilk test. For bivariate comparisons of these continuous scores across demographic groups, the Independent Samples t-test was utilized for normally distributed data, while the Mann-Whitney U test was employed for non-normally distributed data. To ensure the robustness of the findings, a sensitivity analysis was conducted by dichotomizing these scores at the 50th percentile (median) into "Adequate/Inadequate" and "Positive/Negative" categories, as specified in the operational definitions. Bivariate associations for these categorical outcomes and for practice (Yes/No) were assessed using the Chi-square test. Spearman's rank correlation coefficient was used to assess the bivariate relationships between knowledge, attitude, and practice scores.

For the multivariable analysis, variables demonstrating a p-value less than or equal to 0.20 in the bivariate tests were entered into the final models to identify independent associated factors [29]. Multivariable linear regression served as the primary model for identifying associated factors of the continuous knowledge and attitude scores. Conversely, multivariable binary logistic regression was utilized for the primary analysis of practice and for the sensitivity analysis of the dichotomized categories. Assumptions for linear regression, including the normality of residuals and the absence of multicollinearity (assessed via the Variance Inflation Factor), were verified. Statistical significance for all analyses was established at a two-tailed p-value less than 0.05. Post-hoc power analysis of the multivariable logistic regression (blood donation practice; event per variable = 8) yielded 91% power (GPower; OR = 2.0, n = 191, α = 0.05). Given the borderline EPV = 8, sensitivity analysis tested a reduced model excluding father education (p = 0.663 in the main model). Internal consistency of both questionnaires was assessed using Cronbach's α. The knowledge scale showed acceptable reliability (α = 0.73), while the attitude scale had lower but reported reliability (α = 0.43; mean corrected item-total r = 0.21 ± 0.09, range: 0.03–0.35), as shown in the S1 Table in S3 File. EFA confirmed modest attitude scale unidimensionality (KMO = 0.58; Bartlett's p < 0.001; 1 factor, eigenvalue = 1.32 explaining 19% variance; loadings 0.04–0.64), as shown in the S2 Table in S3 File.

## Ethics approval and consent to participate

The study was conducted in accordance with the national ethical guidelines of Nepal, which are aligned with the principles outlined in the Declaration of Helsinki. Ethical approval was obtained from the Institutional Review Committee of the Institute of Medicine (Ref: 449(6-11)E2 078/079), a health research review body under the Nepal Health Research Council, the central authority responsible for implementing ethical standards in health research in Nepal. Participants were provided with an electronic information sheet detailing the study's purpose, procedures, and their right to withdraw at any time. Informed consent was obtained electronically through a click-to-consent mechanism, where participants clicked the "I agree" button to proceed. Participation was entirely voluntary, and the confidentiality of all responses was assured.

## Results

### Demographic characteristics of participants

A total of 191 engineering students participated in the study (33.2% response rate), while 7 students (1.2%) explicitly declined to participate, and the remaining 378 (65.6%) did not respond to the invitation, as shown in Fig 1. The median age of participants was 20 years (IQR = 1 year), ranging from 18 to 22 years. The demographic profile indicates that a majority of the participants were male (73.3%), Hindu (93.2%), and of Brahmin/Chhetri ethnicity (72.3%). Regarding age group, 54.5% of the students were 20 years or older, as shown in Table 1.

The residential and familial background of the cohort was predominantly urban and stable; 57.1% resided in municipalities, and 68.1% lived with both parents. Educational attainment among parents was high, with 76.4% of fathers and 59.7% of mothers having completed secondary education or higher. The sample was nearly equally divided by academic track, with 49.7% enrolled in civil engineering and 50.3% in other engineering disciplines.

### Knowledge regarding blood donation

The assessment of blood donation knowledge revealed high awareness in several fundamental areas, as shown in S3 Table in S3 File. A vast majority of students correctly identified screening as a necessity before donation (89.5%), recognized the types of food to be consumed post-donation (84.3%), and understood that infections can be transmitted through blood (78.0%). Furthermore, knowledge regarding the minimum age (73.3%) and the total duration of the donation process (72.8%) was relatively high.

In contrast, significant knowledge gaps were observed regarding technical and physiological aspects of donation. Only 18.8% of participants were aware that one unit of whole blood can benefit three patients, and only 17.8% correctly identified the minimum duration required between childbirth and donation. Knowledge was also notably low regarding the storage duration of platelets (29.3%), the time required for blood levels to return to normal (29.3%), and the minimum hemoglobin requirement (30.9%).

Regarding legal aspects, 94.2% of participants identified voluntary donation, and 71.2% identified family replacement as legal types. For donation venues, 92.7% and 63.4% correctly identified health centers and community organizations, respectively. Additionally, 84.8% correctly identified that "any company" is not a legal entity, while 48.7% correctly identified that paid donation is not legally permitted.

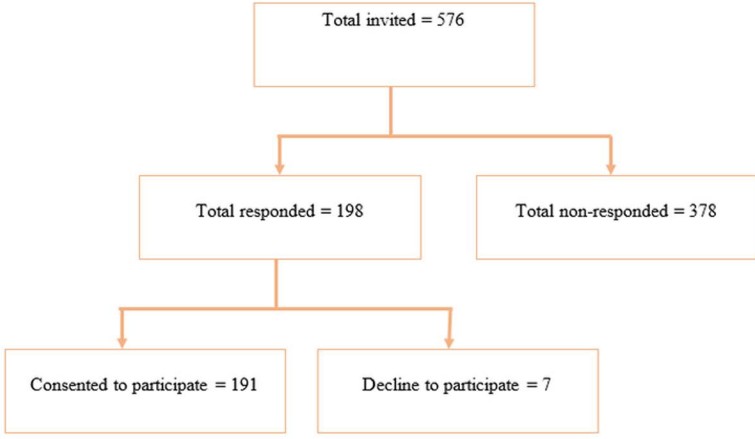

**Fig 1. Flow diagram of participant recruitment and study enrollment.**

**Table 1. Characteristics of first-year engineering students (n = 191).**

| Characteristic | Category | N | % |
|---|---|---|---|
| Age | Less than 20 years | 87 | 45.5 |
| | 20 years or more | 104 | 54.5 |
| Sex | Female | 51 | 26.7 |
| | Male | 140 | 73.3 |
| Religion | Non-Hindu[a] | 13 | 6.8 |
| | Hindu | 178 | 93.2 |
| Ethnicity | Non-Brahmin/Chhetri[b] | 53 | 27.7 |
| | Brahmin/Chhetri | 138 | 72.3 |
| Living with | Non-nuclear[c] | 61 | 31.9 |
| | Both parents | 130 | 68.1 |
| Programme | Other engineering[d] | 96 | 50.3 |
| | Civil engineering | 95 | 49.7 |
| Father's education | Basic or below | 45 | 23.6 |
| | Secondary or above | 146 | 76.4 |
| Mother's education | Basic or below (1–8 years) | 77 | 40.3 |
| | Secondary or above (≥9 years) | 114 | 59.7 |
| Family residency | Municipality | 109 | 57.1 |
| | Metropolitan City | 82 | 42.9 |

Non-Hindu[a] includes Buddhist, Christian, Kirat, and Muslim; Non-Brahmin/Chhetri[b] includes Janjati, Madhesi, Muslim, and Dalit; Non-nuclear[c] includes living alone, with only father, only mother, and other people; Other engineering[d] includes mechanical engineering, Electronics, Communication, & Information Engineering, Computer Engineering, Electrical Engineering, Aerospace Engineering, Chemical Engineering, and Bachelor of Architecture.

## Attitude towards blood donation

Participants reported a high willingness to donate blood, both to relatives (95.8%) and to anyone in need (90.6%). Most respondents (96.3%) viewed blood donation as a noble act, and 97.3% expressed a positive attitude toward the practice. Furthermore, 95.3% would donate regardless of the recipient's religion, and 87.4% identified voluntary donors as the ideal source. Regarding incentives, 77.0% of participants did not expect a reward for donation, as shown in S4 Table in S3 File.

## Practice of blood donation

Regarding practices, 16.8% of participants had previously donated blood, as shown in S5 Table in S3 File. Among these donors (n = 32), the majority had donated once (87.5%), cited voluntary reasons (93.7%), and reported being satisfied (96.9%) with the process. Furthermore, 93.8% of donors expressed a willingness to donate again in the future. Among those who had never donated (n = 159), the primary reasons cited were no specific reason (42.8%), lack of opportunity (34.6%), and fear (20.1%). Additionally, while personal donation rates were low, 68.1% of all participants reported that a family member had donated blood previously.

## Knowledge and attitude

The descriptive statistics for the participants' scores are summarized in Table 2. The mean knowledge score was 13.91 (±3.81), with scores ranging from a minimum of 4 to a maximum of 23. For the attitude domain, participants achieved a median score of 7.00 (IQR: 1.00), with an observed range between 3 and 7.

## Factors associated with knowledge

An independent t-test was conducted to compare knowledge scores across sociodemographic variables, as shown in S6 Table in S3 File. Female students (M = 14.96, SD = 3.69) scored significantly higher than male students (M = 13.53, SD = 3.79; t(189) = 2.33, p = 0.021). The mean difference was 1.43 (95% CI: 0.22 to 2.65), representing a small-to-medium effect size (Cohen's d = 0.38).

No other sociodemographic characteristics, including age (p = 0.408), religion (p = 0.812), ethnicity (p = 0.179), or parental education (Father: p = 0.859; Mother: p = 0.934), showed a statistically significant association with knowledge scores.

A multivariable linear regression was performed to identify associated factors of continuous knowledge scores, as shown in Table 3. The overall model reached borderline statistical significance (F(3, 187) = 2.52, p = 0.060), explaining approximately 4% of the variance (R-squared = 0.04). Sex was the only significant variable in the model; male students scored 1.26 points lower than female students when adjusting for other variables (B = −1.26, 95% CI: −2.51 to −0.01, p = 0.048). Other factors, including ethnicity (p = 0.233) and academic program (p = 0.400), were not significantly associated with knowledge scores. Multicollinearity was not a concern, as all Variance Inflation Factor (VIF) values were below 1.1.

## Factors associated with attitude

Mann-Whitney U tests showed that higher attitude scores were significantly associated with identifying as Hindu (Mean Rank = 98.0 vs. 68.3; U = 797, Z = −2.11, p = 0.035), living with both parents (Mean Rank = 103.1 vs. 80.9; U = 3047, Z = −2.91, p = 0.004), and having a mother with secondary education or higher (Mean Rank = 102.2 vs. 86.9; p = 0.034), as shown in S7 Table in S3 File. No significant differences were observed for age, sex, ethnicity, or residency. Additionally, a weak but significant positive correlation was found between knowledge and attitude scores (rho = .176, p = .015), as shown in S8 Table in S3 File.

In multivariable linear regression, the overall model significantly predicted attitude scores (F(5, 185) = 2.87, p = 0.016, R-squared = 0.07), as shown in Table 4. Knowledge was the only significant independent variable, where a one-unit increase in knowledge was associated with a 0.04-point increase in attitude score (B = 0.04, 95% CI: 0.01 to 0.07, p = 0.025). Identifying as Hindu showed a borderline significant association (p = 0.057), while living arrangements, maternal education, and residency were not significantly associated factors in the adjusted model. Multicollinearity was not a concern, as all Variance Inflation Factor (VIF) values were below 1.2.

Table 2. Descriptive statistics for knowledge and attitude scores (n = 191).

| Variable | Mean (SD) or Median (IQR) | Range |
|---|---|---|
| Knowledge | 13.91 (±3.81) | 4-23 |
| Attitude | 7.00 (1.00) | 3-7 |

Table 3. Linear regression predicting knowledge scores (n = 191).

| Predictor | B | SE | β | t | p-value | 95% CI Lower | 95% CI Upper |
|---|---|---|---|---|---|---|---|
| Male (ref: Female) | −1.26 | 0.63 | −0.15 | −1.99 | 0.048* | −2.51 | −0.01 |
| Hindu (ref: Non-Brahmin/Chhetri) | −0.73 | 0.61 | −0.09 | −1.20 | 0.233 | −1.93 | 0.47 |
| Civil engineering (ref: Other engineering) | −0.47 | 0.56 | −0.06 | −0.84 | 0.400 | −1.57 | 0.63 |

Categorization of comparison groups (Non-Brahmin/Chhetri, and Other engineering) is consistent with the detailed definitions provided in Table 1; *p < .05.

**Table 4. Linear regression predicting attitude scores (n = 191).**

| Predictor | B | SE | β | t | p | 95% CI Lower | 95% CI Upper |
|---|---|---|---|---|---|---|---|
| Hindu (ref: Non-Hindu) | 0.28 | 0.15 | 0.14 | 1.91 | 0.057 | −0.01 | 0.57 |
| Living with both parents (ref: Non-nuclear) | 0.36 | 0.26 | 0.10 | 1.37 | 0.173 | −0.16 | 0.88 |
| Mother's education Secondary (ref: Basic or below) | 0.13 | 0.15 | 0.07 | 0.89 | 0.373 | −0.16 | 0.42 |
| Metropolitan City (ref: Municipality) | 0.08 | 0.14 | 0.04 | 0.53 | 0.598 | −0.21 | 0.36 |
| Knowledge score | 0.04 | 0.02 | 0.16 | 2.26 | 0.025* | 0.01 | 0.07 |

Categorization of comparison groups (Non-Hindu, and Non-nuclear) is consistent with the detailed definitions provided in Table 1; *p < .05.

## Practice and associated factors

Chi-square tests were used to identify sociodemographic factors associated with blood donation history, as shown in S9 Table in S3 File. Age was significantly associated with donation practice; students aged 20 years or older had a higher rate of donation compared to those under 20 (24.0% vs. 8.0%; $\chi^2 = 8.687$, p = 0.003). Sex was also a significant factor, with male students reporting a higher prevalence of donation than female students (20.7% vs. 5.9%; $\chi^2 = 5.896$, p = 0.015). No significant associations were observed for religion, ethnicity, parental education, or residency (p > 0.05), although the academic program showed a marginal association (p = 0.057). Spearman's rank correlation showed that blood donation practice was not significantly associated with either knowledge (rho = .053, p = .466) or attitude (rho = .045, p = .532), as shown in S8 Table in S3 File.

In multivariable logistic regression analysis, age, sex, and academic program were significantly associated factors of blood donation practice, as shown in Table 5. The multivariable logistic regression showed acceptable fit (Hosmer-Lemeshow $\chi^2 = 14.92$, df = 8, p = 0.061), explained 16.3% of variance (Nagelkerke $R^2 = 0.163$), and maintained validity at EPV = 8 (acceptable per Vittinghoff & McCulloch, 2007) [30]. Students aged 20 years or older (OR = 2.83, 95% CI: 1.11–7.22, p = 0.030) and male students (OR = 4.20, 95% CI: 1.16–15.26, p = 0.029) were significantly more likely to have donated blood. Conversely, civil engineering students were 62% less likely to have donated compared to those in other programs (OR = 0.38, 95% CI: 0.16–0.88, p = 0.024). Parental education was not a significantly associated factor in this model.

## Sensitivity analysis

A sensitivity analysis was performed to evaluate the consistency of predictors when treating knowledge as a continuous score versus a binary outcome (≥50th percentile). Findings were sensitive to the method of categorization; while sex was significantly associated with continuous knowledge scores in both bivariate (p = .021) and multivariable linear regression (B = −1.26, p = .048), these associations lost statistical significance in the binary model (p = .083 and p = .260, respectively). All other sociodemographic variables remained consistently non-significant across both analytical approaches. The loss of significance for sex in the categorical model suggests that while gender differences exist in the depth of knowledge,

**Table 5. Logistic regression predicting blood donation practice (n = 191).**

| Predictor | B | SE | Wald | p | OR | 95% CI Lower | 95% CI Upper |
|---|---|---|---|---|---|---|---|
| Age ≥ 20 years (ref: < 20 years) | 1.039 | 0.479 | 4.704 | 0.030* | 2.83 | 1.11 | 7.22 |
| Male (ref: female) | 1.436 | 0.658 | 4.764 | 0.029* | 4.20 | 1.16 | 15.26 |
| Civil eng. (ref: Other engineering) | −0.970 | 0.429 | 5.109 | 0.024* | 0.38 | 0.16 | 0.88 |
| Father: Secondary or above (ref: Basic or below) | −0.197 | 0.453 | 0.190 | 0.663 | 0.82 | 0.34 | 2.00 |

Categorization of comparison groups (Other engineering) is consistent with the detailed definitions provided in Table 1; *p < .05.

they are not robust enough to persist when the data is dichotomized, as shown in S10 and S11 Tables in S3 File. Consequently, the continuous analysis was retained as the primary model to maximize statistical power and precision.

A sensitivity analysis was performed to evaluate the consistency of predictors when treating attitude as a continuous score versus a binary outcome (≥50th percentile). Bivariate analyses yielded identical significant factors across both methods: religion (p = .035 vs. p = .047), living arrangement (p = .004 vs. p = .002), and mother's education (p = .034 vs. p = .039). Similarly, the positive correlation between knowledge and attitude remained significant in both continuous (rho = .176, p = .015) and categorical (rho = .153, p = .034) formats. In multivariable modeling, knowledge remained a stable independent associated factor of attitude in both the linear (B = 0.04, p = .025) and logistic (OR = 1.09, p = .036) regressions. While living with both parents reached significance only in the binary model (OR = 2.40, p = .009), the overall high level of agreement between the models confirms the robustness of the primary continuous analysis, as shown in S12–S14 Tables in S3 File. For the practice of blood donation, the reduced model (age, sex, program; EPV = 10.7) confirmed the robustness of the main findings: superior fit (Hosmer-Lemeshow $\chi^2$ = 6.31, df = 6, p = 0.390) and equivalent explanatory power (Nagelkerke $R^2$ = 0.161 vs 0.163), as shown in S15 Table in S3 File.

## Discussion

This study provides the first comprehensive assessment of blood donation knowledge, attitudes, and practices (KAP) among engineering students in Nepal. Our findings reveal a population with high pro-donation attitudes but relatively moderate knowledge and low actual practice. Donation behavior was not linked to knowledge or attitude scores, but was associated with demographic and academic factors such as age, sex, and specific engineering program. This descriptive "knowledge-practice gap" suggests that while students are ideologically willing to donate, structural or personal barriers—rather than a lack of awareness—may be hindering actual donation.

The mean knowledge score in this study (13.91 ± 3.81), representing 60.4% of the total score, was just above the 50% threshold, falling short of the 75% level often considered representative of "good knowledge" necessary for sustainable behavior change [31]. This gap is significant because, while knowledge alone does not guarantee practice, it remains a critical motivator and a prerequisite for fostering a self-sustaining pool of voluntary donors. This shortfall may stem from the lack of formal emphasis on blood donation in school-level curricula, despite its critical role in medical treatment. Our findings align with a study among school students in India, which reported a similar knowledge prevalence of 57.1% [19]. In contrast, a study among Ethiopian healthcare workers found a significantly higher prevalence of 82.6% [25]. This disparity is likely due to the professional exposure of healthcare workers, whereas engineering students rely primarily on school-level health education. A specific knowledge deficit was observed regarding the impact of donation; only 18.8% of participants correctly identified that a single donation can benefit up to three patients. Highlighting this "multiplier effect" might significantly enhance donation rates by appealing to students' sense of social responsibility. Conversely, awareness of the minimum donation age was relatively high (73.3%), surpassing the 54.5% reported among undergraduate health science students in Ethiopia [21]. This may be attributed to our study site's status as a central hub of Tribhuvan University, which facilitates robust student networks and frequent campus-based social activities that normalize the logistical aspects of donation.

Attitude serves as a crucial mediator between knowledge and practice; while knowledge provides the factual basis, a positive attitude enables the behavioral transition to donation. In this study, the median attitude score was 7.00 (IQR: 1.00), with 57.1% of participants categorized as having a "positive" attitude based on the 50th percentile. This proportion aligns with research among Ethiopian healthcare professionals, which reported a similar prevalence (58.7%) [25]. The overwhelming optimism and willingness to donate suggest that, despite technical knowledge gaps, students maintain a highly favorable perception of donation. However, the high median score may partially reflect "social desirability bias," where participants provide morally "correct" answers until confronted with the physical reality of donation. This is a common phenomenon in blood donation research across diverse cultural contexts, including Ethiopia and Saudi Arabia, where donation is almost universally viewed as a noble and moral act [21,25,32].

 

Interestingly, nearly one-fourth of participants expected some form of reward for their contribution. While voluntary non-remunerated donation is the global gold standard, expectations for transportation reimbursement, refreshments, or formal recognition are frequently documented [33,34]. This trend mirrors findings from Saudi Arabia, where donors often anticipate tangible incentives [32]. Addressing these expectations—perhaps by emphasizing "symbolic" non-monetary rewards—might be key to bridging the gap between high ideological support and low actual donation rates.

The prevalence of blood donation in this study was 16.8%. This finding is consistent with research among similar student populations [35,36], yet falls below the higher rates reported in several other academic settings [17,19,28,37–43]. Conversely, our results exceed the donation practices documented in certain developing contexts, such as among health science students in Ethiopia [21,44]. These cross-national discrepancies likely stem from variations in cultural norms, public awareness levels, and the maturity of the local health infrastructure.

Notably, over 93% of the donors in our cohort were voluntary, non-remunerated donors who reported high satisfaction and a willingness to return. This aligns perfectly with the WHO global framework for achieving 100% voluntary repeat donations [9]. However, for the majority who had never donated, the primary barriers were "no specific reason" and a "perceived lack of opportunity," followed by fear. These barriers—consistent with findings in other undergraduate populations [28,35–37]—suggest that the low donation rate is not due to active opposition, but rather a lack of structural mobilization. This represents a significant public health opportunity: implementing targeted on-campus blood drives and addressing "fear-based" misconceptions could easily convert these "willing non-donors" into active contributors.

Multivariable analysis revealed distinct socio-demographic factors across the three domains of knowledge, attitude, and practice. Regarding knowledge, male students scored significantly lower than their female counterparts. While some studies among medical students report no gender-based knowledge gap [43], our findings align with research from Gaza, where gender is significantly linked with awareness levels [35]. This suggests that female engineering students may be more proactive in seeking health-related information or have higher exposure to altruistic social networks.

Knowledge score was the sole significant factor correlated with attitude, indicating a statistically significant but very small correlation. A one-unit knowledge increase is associated with only a 0.04-point attitude improvement, suggesting minimal practical impact. This relationship is supported by behavioral learning theories and similar findings among Chinese college students [42,45], reinforcing the idea that educational interventions are a prerequisite for shifting student mindsets.

However, the most striking results appeared in the prediction of actual donation behavior. The knowledge and attitude were not the significant associated factors, but age and sex remained powerful associated factors. Students aged 20 years or older were nearly three times more likely to have donated, a finding corroborated by studies in Ethiopia, where maturity and increased campus residency time provide more donation opportunities [44]. Furthermore, male students were four times more likely to be donors despite having lower knowledge scores than females. This paradox—where higher knowledge in females does not translate to practice—may be attributed to physiological barriers such as lower body weight, higher prevalence of anemia, or fear of needles, institutional factors such as heavy engineering workload, lack of organization of blood donation camps, and cultural factors such as misconception and lack of personal request made for the blood donation which are frequently reported obstacles for female donors in South Asia and in the current study [28,35–37]. Interestingly, civil engineering students were significantly less likely to donate than those in other engineering disciplines. This may suggest that academic workload or different social subcultures within engineering departments may influence behavioral participation, warranting further investigation into program-specific barriers.

## Implications

The findings of this study offer several evidence-based implications for stakeholders and authorities governing blood transfusion services in Nepal. Primarily, the delineation of current knowledge and practice levels provides a necessary empirical foundation for planners to navigate recruitment strategies. Given that "no specific reason" and "no opportunity"

were the most frequently cited barriers to donation, there is a clear imperative to move beyond general awareness toward structural mobilization.

To address the perceived lack of opportunity, academic administrations should coordinate with regional blood centers to organize periodic, on-campus donation campaigns. Establishing on-site facilities not only eliminates logistical hurdles but also encourages the transition from first-time to repeat donor status among willing students. Furthermore, the development of peer-led advocacy through dedicated student clubs or existing campus chapters is essential. These student-driven initiatives can host awareness sessions and World Blood Donor Day celebrations, which serve to dispel physiological fears and cultivate a sense of civic responsibility and social welfare.

Regarding incentives, while maintaining the global standard of voluntary non-remunerated donation, the implementation of formal recognition awards, provision of refreshments, and reimbursement of transportation costs for off-site events may significantly bolster student readiness to participate. Ultimately, these local efforts must be supported by the effective implementation of Nepal's National Blood Transfusion Policy (2017), which emphasizes the need for multisectoral collaboration to strengthen the national donor pool [46]. By aligning campus-level initiatives with this national framework, stakeholders can transform a willing engineering student population into a reliable and informed donor advocate community.

### Limitations and strengths

This study has several limitations that warrant consideration. First, the cross-sectional design precludes the inference of causal relationships between the identified predictors and donation outcomes. Second, while focusing on first-year engineering students provides a unique demographic insight, the results from a single institution may not be fully generalizable to all undergraduate populations in Nepal. Third, the response rate (33.2%) introduces a risk of selection bias, as students with a pre-existing interest in blood donation may have been more likely to participate. Fourth, non-respondent demographics are unavailable (email-only access), limiting population-sample discrepancy exploration. Female over-representation (26.7% vs. national ~17%) suggests response-propensity bias rather than population composition; it limits gender generalizability [47]. Fifth, the internal consistency of our assessment tools in the final study population showed variance; while the knowledge scale demonstrated respectable reliability (Cronbach's alpha = 0.73), the attitude scale showed lower internal consistency (Cronbach's alpha = 0.43). This lower alpha for the attitude domain is likely a reflection of the significant "ceiling effect"—where the high homogeneity of positive responses reduced the scale's variance—suggesting that attitude measurements may be subject to social desirability bias. Furthermore, the web-based nature of the survey may have limited participation from students with inconsistent internet connectivity.

Despite these constraints, the study possesses several notable strengths. It is, to our knowledge, the first to specifically evaluate the blood donation KAP of engineering students in Nepal. To address the measurement challenges and the skewness in attitude data, we moved beyond simple categorization and utilized continuous scores as the primary outcome in our multivariable models. This methodological choice preserved statistical power and allowed for a more rigorous analysis of how knowledge levels and demographic factors independently drive donation practice. Additionally, the high percentage of voluntary, non-remunerated donors (over 93%) among the participant pool provides a high-quality baseline for future longitudinal research and targeted recruitment strategies.

### Conclusion

This study reveals that while engineering students in Nepal maintain moderate knowledge and highly favorable attitudes toward blood donation, actual practice remains low. Our findings highlight a descriptive "knowledge-practice gap," where behavior is primarily associated with age, sex, and academic program rather than awareness alone. Notably, despite higher knowledge among females, male students were four times more likely to donate, suggesting the presence of gender-specific physiological or structural barriers. Given that "lack of opportunity" was a primary deterrent, interventions should shift from general awareness to structural mobilization, specifically through regular on-campus donation drives and

the integration of blood donation into formal curricula. Strengthening institutional coordination in line with Nepal's national policy is essential to transform this willing student population into a reliable, self-sustaining donor pool.

## Supporting information

**S1 File. STROBE statement checklist.**
(DOCX)

**S2 File. Study questionnaire.**
(DOCX)

**S3 File. Supplementary tables.**
(DOCX)

**S4 File. Minimal data set.**
(XLSX)

## Acknowledgments

We extend our sincere appreciation to the first-year students of Pulchowk Engineering Campus for their cooperation and for sharing the valuable information that formed the basis of this research; this study would not have been possible without their voluntary participation. We are also grateful to the authorities of Pulchowk Engineering Campus for their institutional support and cooperation during the data collection process. Finally, we would like to express our gratitude to the faculty and leadership of the Central Department of Public Health for their academic guidance throughout the duration of this study.

## Author contributions

**Conceptualization:** Bhola Teli, Durga Rijal, Sulochan GC, Ashok Khanal, Anil Kumar Singh.

**Data curation:** Bhola Teli.

**Formal analysis:** Bhola Teli, Anil Kumar Singh.

**Funding acquisition:** Bhola Teli.

**Investigation:** Bhola Teli.

**Methodology:** Bhola Teli, Sulochan GC, Anil Kumar Singh.

**Project administration:** Bhola Teli.

**Resources:** Bhola Teli.

**Software:** Bhola Teli.

**Supervision:** Bhola Teli, Ashok Khanal, Anil Kumar Singh.

**Validation:** Bhola Teli, Ashok Khanal.

**Visualization:** Bhola Teli.

**Writing – original draft:** Bhola Teli.

**Writing – review & editing:** Bhola Teli, Durga Rijal, Sulochan GC, Ashok Khanal, Anil Kumar Singh.

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
