## [Decision Letter · Decision Letter 0]

8 Mar 2026

PONE-D-26-05598Knowledge, Attitude, and Practice on Blood Donation among Undergraduate First-Year Engineering Students in NepalPLOS One

Dear Dr. Teli,

Thank you for submitting your manuscript to PLOS ONE. After careful consideration, we feel that it has merit but does not fully meet PLOS ONE’s publication criteria as it currently stands. Therefore, we invite you to submit a revised version of the manuscript that addresses the points raised during the review process.

We look forward to receiving your revised manuscript.

Kind regards,

Nasar Alwahaibi, PhD

Academic Editor

PLOS One

Journal Requirements:

“The authors received no specific grant from any funding agency in the public, commercial, or not for-profit sectors for this research.”

3. Please remove your figures from within your manuscript file, leaving only the individual TIFF/EPS image files, uploaded separately. These will be automatically included in the reviewers’ PDF.

4. We note that there is identifying data in the Supporting Information file “S4 File. Minimal Data Set..xlsx”. Due to the inclusion of these potentially identifying data, we have removed this file from your file inventory. Prior to sharing human research participant data, authors should consult with an ethics committee to ensure data are shared in accordance with participant consent and all applicable local laws.

-Location data

Please remove or anonymize all personal information (<Age>), ensure that the data shared are in accordance with participant consent, and re-upload a fully anonymized data set. Please note that spreadsheet columns with personal information must be removed and not hidden as all hidden columns will appear in the published file.

Reviewers' comments:

Reviewer's Responses to Questions

**Comments to the Author**

1. Is the manuscript technically sound, and do the data support the conclusions?

Reviewer #1: Yes

Reviewer #2: Yes

2. Has the statistical analysis been performed appropriately and rigorously? 

Reviewer #1: No

Reviewer #2: Yes

3. Have the authors made all data underlying the findings in their manuscript fully available?

Reviewer #1: Yes

Reviewer #2: Yes

4. Is the manuscript presented in an intelligible fashion and written in standard English?

Reviewer #1: No

Reviewer #2: Yes

5. Review Comments to the Author

Reviewer #1: Thank you, editorial team, inviting me to review this important topic. The topic is relevant to public health and transfusion medicine, particularly in low- and middle-income settings where voluntary blood donation remains insufficient. The study addresses an understudied population (engineering students), which adds novelty within the Nepalese context. The manuscript is generally well-structured and clearly written. However, there are several methodological and analytical concerns that require substantial clarification and revision before the manuscript can be considered for publication. The most critical issues relate to:

• Low response rate and potential selection bias

• Limited number of outcome events in logistic regression (risk of overfitting)

• Low internal consistency of the attitude scale

• Interpretation of regression findings and causal language

Major Comments

1. Response Rate and Potential Selection Bias

The response rate was 33.2% (191/576), substantially lower than the calculated target sample size (451). This raises concerns about:

• Non-response bias

• Representativeness of the study sample

• Generalizability of findings

Required clarifications:

1. Were demographic characteristics of respondents compared with the full eligible population?

2. Was any assessment of non-response bias conducted?

3. How does the lower-than-planned sample size affect statistical power and model stability?

The limitations section acknowledges response rate but does not sufficiently discuss implications for internal validity.

2. Sample Size and Logistic Regression Stability

Only 32 participants reported prior blood donation (16.8%). The logistic regression model includes multiple predictors (age, sex, academic program, father’s education).

With only 32 events:

• The events-per-variable (EPV) ratio may be <10.

• Risk of model overfitting is substantial.

• Wide confidence intervals (e.g., OR = 4.20; 95% CI: 1.16–15.26) indicate imprecision.

Required clarifications:

1. Was EPV formally assessed?

2. Were alternative modelling strategies considered (e.g., reduced models)?

3. Was model goodness-of-fit evaluated (e.g., Hosmer–Lemeshow test)?

The stability and reliability of the logistic regression findings must be better justified.

3. Internal Consistency of Attitude Scale (Cronbach’s α = 0.43)

The reported Cronbach’s alpha for the attitude scale is 0.43, indicating poor internal consistency.

Although the authors attribute this to a ceiling effect, an alpha below 0.5 raises concerns regarding:

• Construct validity

• Use of a composite continuous score in regression

• Interpretation of subtle differences in attitude

Required clarifications:

1. Were item-total correlations assessed?

2. Was exploratory factor analysis considered?

3. How do the authors justify using this scale in multivariable linear regression?

This issue significantly affects the robustness of the attitude-related findings.

4. Interpretation of Regression Models

4.1 Knowledge Model

The overall model is borderline non-significant (F(3,187) = 2.52, p = 0.060), with R² = 0.04.

• The model explains only 4% of the variance.

• Despite this, sex is interpreted as a meaningful predictor.

Please clarify how findings from a model with limited explanatory power should be interpreted.

4.2 Attitude Model

R² = 0.07. Knowledge is statistically significant (B = 0.04), but the effect size is very small.

The practical significance of this coefficient should be discussed more cautiously.

4.3 Knowledge–Practice Gap

The manuscript repeatedly emphasizes a “knowledge-practice gap.” However:

• Knowledge was not significantly associated with practice.

• Attitude was also not associated with practice.

The conclusion appears to infer a conceptual gap rather than empirically demonstrated mediation.

Please clarify:

1. How is the “knowledge-practice gap” operationally defined?

2. Is it descriptive (high attitude, low practice) rather than analytic?

This framing should be refined.

5. Causal Language in Discussion

Several sections imply causality (e.g., “knowledge drives attitude”).

Given the cross-sectional design, all associations must be interpreted cautiously and described as associative, not causal.

Please revise language throughout to avoid causal implications.

6. Dichotomization of Variables

Age was dichotomized (<20 vs. ≥20), and knowledge and attitude were categorized at the 50th percentile for sensitivity analysis.

Dichotomization:

• Reduces statistical power

• May obscure meaningful gradients

• Appears arbitrary

Please justify:

1. Why age was not treated continuously

2. Why median splits were chosen rather than evidence-based thresholds

7. Social Desirability and Ceiling Effect

Attitude scores are extremely skewed:

• Median = 7 (maximum = 7)

• 97% positive responses

This strongly suggests social desirability bias.

The manuscript acknowledges this but does not sufficiently explore its impact on:

• Attitude reliability

• Limited variability

• Interpretation of regression results

Further elaboration is needed.

Minor Comments

1. The physiological description of blood in the Introduction could be condensed to maintain focus.

2. Please confirm whether more recent WHO data are available beyond 2018.

3. Clarify whether the male proportion (73.3%) reflects campus enrolment distribution.

4. Some speculative explanations (e.g., academic workload differences between programs) should be clearly labelled as hypotheses.

5. Please ensure consistency in terminology (e.g., “knowledge-practice gap” vs. “knowledge–practice gap”).

6. Consider reporting model fit statistics for logistic regression.

Good luck!

Reviewer TA

Reviewer #2: This manuscript presents a cross-sectional study assessing knowledge, attitudes, and practices (KAP) regarding blood donation among first-year engineering students in Nepal. The topic is relevant to public health, particularly in low- and middle-income countries where maintaining a sufficient voluntary blood supply remains challenging. The authors address an understudied population (engineering students), which adds some novelty to the existing literature.

The manuscript is generally well structured and follows standard reporting elements for observational studies. The use of multivariable regression models to examine predictors of knowledge, attitude, and practice is appropriate. The discussion attempts to contextualize findings within global literature on blood donation behaviors.

However, several methodological, reporting, and clarity issues require attention before the manuscript can be considered for publication. These mainly relate to questionnaire validation, sampling bias, statistical interpretation, and clarity of the analysis strategy. Addressing these concerns will improve the scientific rigor and transparency of the study.

1. Questionnaire Development and Validation

The manuscript states that the questionnaire was developed based on literature review and expert consultation and pretested among public health students. However, important details regarding instrument validation are missing.

Specifically:

The process of content validation is not described (e.g., number of experts involved, evaluation method).

Construct validity was not assessed.

Reliability results are insufficiently reported. Although Cronbach’s alpha is mentioned, only values in the limitations section are provided.

Given that the attitude scale has a very low internal consistency (α = 0.43), the authors should:

Report reliability results clearly in the Methods or Results section.

Discuss the implications of this low reliability on interpretation of the attitude findings.

Consider whether the scale should be revised or interpreted cautiously.

2. Sampling Strategy and Response Bias

The study invited the entire population of 576 students, but the final response rate was 33.2%. This relatively low response rate introduces a potential non-response bias, which may affect the representativeness of the findings.

The authors should:

Discuss how non-response bias might influence the results.

Clarify whether respondents differed from non-respondents in available demographic characteristics.

Provide justification for the assumption that the sample remains representative of the student population.

3. Sample Size Calculation and Study Power

The sample size calculation initially estimated a minimum of 176 participants, yet the final response was 191 participants. However, the manuscript also states that the adjusted sample size for non-response was 451, which was not achieved.

The authors should clarify:

Whether the study remained adequately powered for the planned analyses.

Why the adjusted sample size target was not reached.

Whether a post-hoc power analysis was considered.

4. Statistical Analysis and Model Interpretation

The statistical analysis approach is generally appropriate but requires clarification in several areas.

Concerns include:

The linear regression model for knowledge explains only 4% of the variance (R² = 0.04) and the model significance is borderline (p = 0.060), yet the discussion treats the predictor as a strong determinant. Interpretation should be more cautious.

The authors use continuous scores for the primary model and dichotomized variables for sensitivity analysis, which is acceptable; however, the rationale should be explained more clearly earlier in the Methods section.

The selection criterion for variables entering multivariable models (p ≤ 0.20) should be justified with appropriate references.

Potential multicollinearity diagnostics are mentioned but the results should be reported explicitly.

5. Interpretation of Gender Differences

The manuscript reports that female students had higher knowledge scores but were less likely to donate blood.

While the discussion proposes physiological explanations (e.g., anemia, body weight), these factors were not measured in the study. Therefore, these interpretations should be presented as hypotheses rather than conclusions.

The discussion should emphasize that the gender differences may also be influenced by cultural, psychological, or institutional factors.

6. Overly Lengthy Introduction and Discussion

The introduction contains extensive background information about blood composition and donation types that is not directly relevant to the study objectives. Similarly, parts of the discussion are somewhat repetitive and could be condensed.

The manuscript would benefit from:

Shortening the introductory biological description

Focusing more on existing KAP literature and research gaps

6. PLOS authors have the option to publish the peer review history of their article (what does this mean?). If published, this will include your full peer review and any attached files.

Reviewer #1: **Yes:**Temesgen Anjulo Ageru

Reviewer #2: **Yes:**Beesan Maraqa

---

## [Author Response · Author response to Decision Letter 1]

13 Apr 2026

All reviewer comments and journal requirements have been addressed point-by-point in the uploaded 'Response to Reviewers' file.

---

## [Decision Letter · Decision Letter 1]

27 Apr 2026

Knowledge, Attitude, and Practice on Blood Donation among Undergraduate First-Year Engineering Students in Nepal

PONE-D-26-05598R1

Dear Dr. Teli,

We’re pleased to inform you that your manuscript has been judged scientifically suitable for publication and will be formally accepted for publication once it meets all outstanding technical requirements.

Kind regards,

Nasar Alwahaibi, PhD

Academic Editor

PLOS One

Additional Editor Comments (optional):

Reviewers' comments:

Reviewer's Responses to Questions

**Comments to the Author**

1. If the authors have adequately addressed your comments raised in a previous round of review and you feel that this manuscript is now acceptable for publication, you may indicate that here to bypass the “Comments to the Author” section, enter your conflict of interest statement in the “Confidential to Editor” section, and submit your "Accept" recommendation.

Reviewer #1: All comments have been addressed

2. Is the manuscript technically sound, and do the data support the conclusions?

Reviewer #1: Yes

3. Has the statistical analysis been performed appropriately and rigorously? 

Reviewer #1: Yes

4. Have the authors made all data underlying the findings in their manuscript fully available?

Reviewer #1: Yes

5. Is the manuscript presented in an intelligible fashion and written in standard English?

Reviewer #1: Yes

6. Review Comments to the Author

Reviewer #1: The authors have provided thorough, point-by-point responses with appropriate manuscript revisions. All methodological concerns have been transparently addressed, limitations are clearly stated, and interpretations are now appropriately cautious.

Well done to the authors for a responsive and rigorous revision.

7. PLOS authors have the option to publish the peer review history of their article (what does this mean?). If published, this will include your full peer review and any attached files.

Reviewer #1: **Yes:**Dr. Temesgen Anjulo Ageru

---

## [Editor Report · Acceptance letter]

PONE-D-26-05598R1

PLOS One

Dear Dr. Teli,

I'm pleased to inform you that your manuscript has been deemed suitable for publication in PLOS One. Congratulations! Your manuscript is now being handed over to our production team.

Kind regards,

on behalf of

Dr. Nasar Alwahaibi

Academic Editor

PLOS One